# The Outcomes of an Opioid-Free Anaesthetic Plan in Fourteen Dogs Undergoing Enucleation Using an Ultrasound-Guided Supra-Temporal Retrobulbar Block: A Retrospective Case Series

**DOI:** 10.3390/ani13132059

**Published:** 2023-06-22

**Authors:** Gerardo Citarella, Daniele Corona, Eamonn Parsons, Stamatina Giannikaki, Eva Rioja

**Affiliations:** 1Section of Anaesthesiology, Department of Clinical Diagnostics and Services, Vetsuisse Faculty, University of Zurich, 8057 Zurich, Switzerland; daniele.corona@uzh.ch; 2Optivet Referrals, Ltd., Havant, PO9 2NJ, UK; eamonnparsons@gmail.com (E.P.); eva@optivet.com (E.R.); 3Meyevet, 10431 Athens, Greece; sgiannikaki@yahoo.gr

**Keywords:** retrobulbar-block, enucleation, opioid-free anaesthesia

## Abstract

**Simple Summary:**

In recent years, loco-regional anesthesia has become a common practice in both human and veterinary medicine. In particular, it is widely used in the treatment of perioperative analgesia for neurological and orthopedic surgery. The advantage of successfully applying loco-regional anesthesia is better perioperative pain relief with a lesser need for systemic analgesic drugs. This case series aims to show how the practice of loco-regional anesthesia (retrobulbar block) has allowed good management of perioperative analgesia without the use of systemic opioids in 13 out of 14 dogs. A multimodal approach to perioperative analgesia focused on the use of a retrobulbar ultrasound-guided block with a supra-temporal approach was used in 14 dogs undergoing elective unilateral enucleation. A total of 2 out of 14 patients received rescue analgesia intraoperatively, while no patients required rescue analgesia during the postoperative period. No complications were recorded during the perioperative period, and all patients were discharged within the same day.

**Abstract:**

The objective of this retrospective case series is to report the outcomes of an opioid-free anesthetic plan in dogs undergoing enucleation surgery. A total of 14 dogs were admitted for enucleation between March and December 2020. A multimodal approach to perioperative analgesia was used, with a focus on retrobulbar anesthesia. A combination of an ultrasound-guided retrobulbblock with a supratemporal approach in association with ketamine, dexmedetomidine and non-steroidal anti-inflammatory drugs was used in the reported cases. Intraoperative nociception was defined as an increase of 20% from the baseline in one or more of the following parameters: heart rate, respiratory rate or mean arterial pressure. An ultrasound-guided retrobulbar block in an opioid-free anesthesia regime was effective at managing the perioperative analgesia of 13 out of 14 dogs. In only one case, a bolus of fentanyl was administered to treat intraoperative nociception. Recovery was uneventful in all the dogs, and the postoperative pain scores remained below the intervention threshold at all time points. To the authors’ knowledge, this is the first reported case series of opioid-free anesthesia for enucleation in dogs.

## 1. Introduction

The first use of opioids in clinical practice was reported in 1853, when Dr. Alexander Wood performed a subcutaneous injection of morphine [1]. Afterward, with the idea of “balanced anesthesia” described by Lundy in 1931 [2], opioids became an essential medication for the management of acute surgical pain in human and veterinary medicine [3,4].

Opioids have high clinical efficacy, are safe when administered at licensed doses in dogs and cats. Moreover, they have the benefit of being antagonizable [5]. The main opioid receptors are located in the central and peripheral nervous systems. The binding of these drugs to opioid receptors causes a conformational change that increases their affinity for guanine nucleotide binding proteins called G-coupled proteins [5]. These mechanisms decrease the release of excitatory neurotransmitters and hyperpolarize the neuronal membrane, thereby reducing the painful stimuli from nociceptive activation without interfering with proprioception [6]. In small animal medicine, opioids are routinely administered to produce analgesia and sedation as part of pre-anesthetic medication. It is also possible to administer opioids as bolus or constant rate infusion (CRI) during surgery to provide intraoperative analgesia. Moreover, their action can lead to a significant inhalant anesthetic-sparing effect [7,8]. The latter may result in an improvement in the cardiovascular function, leading to greater hemodynamic stability during surgery [9].

For these reasons, veterinarians feel comfortable with the use of opioids, and the adoption of opioid-free anesthesia (OFA) regimens constitutes a difficult challenge due to the belief that intraoperative analgesia will not be sufficient if no opioid is administered [10]. However, the use of opioids is not without complications. Some of the most common side effects related to opioid administration include nausea, reduced gastrointestinal motility, ileus, panting, respiratory depression, opioid tolerance, opioid-induced hyperalgesia (OIH), urinary retention, and negative immunomodulatory effects [11]. These effects can be more or less pronounced, depending on the molecule, dosage, species, route, and rate of administration.

Opioids are Schedule 2 controlled drugs in the UK (Misuse of Drugs Regulations, 2001) and are subject to numerous restrictions in some parts of the world.

In fact, in the United States, due to the emerging opioid crisis, there has been a major shortage of opioid production in recent years, which has also spilled over into the veterinary market, especially with regard to the production of full mu agonists [12].

For these reasons, OFA represents an emergent field in both human and veterinary medicine, drawing huge interest because of the possibility to reduce opioid use. In order to perform OFA and provide adequate analgesia, it is essential to adopt a multimodal approach to perioperative analgesia [13,14]. Combinations of loco-regional anesthetic techniques with non-steroidal anti-inflammatory drugs (NSAIDs), alpha 2 adrenergic agonists, ketamine, and paracetamol are the common approaches used in OFA to ensure an adequate level of intra- and postoperative analgesia [12].

Removal of the globe is usually performed in end-stage ocular disease, often resulting in blind and painful eyes refractory to topical treatment. Enucleation, despite being a routine procedure, represents a painful operation in which the adoption of loco-regional anesthesia techniques is strongly recommended by the authors.

Periocular anesthesia can be performed by means of retrobulbar or peribulbar nerve blocks, performed blindly or under ultrasound guidance, and with different approaches to needle insertion.

The ultimate aim of periocular regional anesthesia is to desensitize the ophthalmic nerve and its branches, which provide the main sensory supply to the globe and surrounding tissues [15,16].

The objective of this retrospective case series is to describe the outcomes of 14 cases undergoing unilateral enucleation where an opioid-free anesthetic plan was selected. The analgesic approach included an ultrasound (US)-guided retrobulbar block (RBB) using a supra-temporal approach in combination with dexmedetomidine, ketamine, or both.

## 2. Methodology

### 2.1. Cases Presentation

Fourteen dogs (5 males and 9 females) of various breeds, with an American Society of Anesthesiologists (ASA) physical status of category II and with a median (range) weight of 17.8 (4.2–53) kg and a median age of 7.5 (1–13) years old, were referred for unilateral enucleation surgery to Optivet Referrals during a 10-month period from March to December 2020. The cases were numbered from 1 to 14, and all medical details are reported in Table 1.

The physical examinations, other than the ocular diseases, were unremarkable for all dogs, and serum biochemistry and hematology tests were performed if deemed necessary based on age or comorbidities.

Unilateral enucleation was selected and performed due to end-stage chronic painful disease in the affected eye.

### 2.2. Treatment

On the day of surgery, food was withheld for at least 8 h, with water being offered up until the time of pre-anesthetic medication and receiving omeprazole (1 mg/kg per os (PO)) upon admission to the hospital. Pre-medication was administered intramuscularly (IM) in 12 dogs and intravenously (IV) in 2 cases. General anesthesia was performed by anesthesia interns and nurses. The RBB was performed by the same anesthesia intern (GC) under the supervision of a board-certified American and European anesthesia specialist (ERG). The anesthetic protocol used for each case is described in detail in Table 1.
animals-13-02059-t001_Table 1Table 1Patient details, anesthetic protocol information, and postoperative pain score.
Case 1 12 Years Jack Russell Terrier, FN, 5.6 kgCase 2 11 Years Jack Russel Terrier, M, 7.4 kgCase 3 7 Years Alaskan Malamute, FN, 35 kgCase 4 10 Years Cocker Spaniel, FN, 13.8 kgCase 5 11 Years Shih Tzu X, FN, 12.6 kgCase 6 6 Years Chihuahua, M, 4.2 kgCase 7 7 Years Great Dane, FN, 53 kgCase 8 13 Years Bearded Collie, FN, 22.9 kgCase 9 1 Year Bull Mastiff, F, 49.5 kgCase 10 12 Years English Springer Spaniel, MN, 21.8 kgCase 11 6 Years Malti-Poo, MN, 7.3 kgCase 12 8 Years Bassett, FN, 29.3 kgCase 13 8 Years Maltese Terrier X, FN, 7.6 kgCase 14 7 Years Springer X, M, 27.2 kg**Procedure**Enucleation ODEnucleation OSEnucleation ODEnucleation OSEnucleation OSEnucleation ODEnucleation OSEnucleation OSEnucleation OSEnucleation OSEnucleation ODEnucleation OSEnucleation ODEnucleation OD**Current medication**OmeprazoleBrinzolamide/Timolol OmeprazoleParacetamol (10 mg/kg) PO OmeprazoleOmeprazoleOmeprazoleOmeprazoleDorzolamide/ Timolol OmeprazoleMeloxicam PO Maropitant (1 mg/kg) IV OmeprazoleOmeprazoleMeloxicam PO OmeprazoleOmeprazoleOmeprazolePhenobarbitone Levetiracetam OmeprazoleMeloxicam PO Omeprazole**Premedication**Medetomidine (0.01 mg/kg) IM Acepromazine (0.005 mg/kg) IMMedetomidine (0.008 mg/kg) IM Acepromazine (0.005 mg/kg) IMAcepromazine (0.005 mg/kg) IM Medetomidine (0.006 mg/kg) IMDexmedetomidine (0.001 mg/kg) IV Acepromazine (0.02 mg/kg) IVDexmedetomidine (0.001 mg/kg) IV Acepromazine (0.005 mg/kg) IVMedetomidine (0.006 mg/kg) IMDexmedetomidine (0.001 mg/kg) IVAcepromazine (0.005 mg/kg) IVDexmedetomidine (0.005 mg/kg) IMAcepromazine (0.015 mg/kg) IVDexmedetomidine (0.005 mg/kg) IM Acepromazine (0.01 mg/kg) IMDexmedetomidine (0.005 mg/kg) IM Followed by another top up of (0.005 mg/kg) Dexmedetomidine IMDexmedetomidine (0.005 mg/kg) IM Acepromazine (0.01 mg/kg) IMMedetomidine (0.01 mg/kg) IM Acepromazine (0.01 mg/kg) IM**Induction**Alfaxalone (1 mg/kg) IVAlfaxalone (1.5 mg/kg) IVPropofol (2 mg/kg) IV Ketamine (1 mg/kg) IVAlfaxalone (1.5 mg/kg) IVAlfaxalone (1.5 mg/kg) IVAlfaxalone (1.5 mg/kg) IVPropofol (2 mg/kg) IV Ketamine (1 mg/kg) IVPropofol (2 mg/kg) IV Ketamine (1 mg/kg) IVPropofol (1.5 mg/kg) IV Ketamine (1 mg/kg) IVAlfaxalone (1 mg/kg) IV Midazolam (0.3 mg/kg) IVAlfaxalone (1 mg/kg) IVPropofol (1 mg/kg) IV Ketamine (1 mg/kg) IVPropofol (2 mg/kg) IVAlfaxalone (1 mg/kg) IV**Maintenance**Sevoflurane delivered in 100% oxygenSevoflurane delivered in 100% oxygenSevoflurane delivered in 100% oxygenSevoflurane delivered in 100% oxygenSevoflurane delivered in 100% oxygenSevoflurane delivered in 100% oxygenSevoflurane delivered in 100% oxygenSevoflurane delivered in 100% oxygenSevoflurane delivered in 100% oxygenSevoflurane delivered in 100% oxygenSevoflurane delivered in 100% oxygenSevoflurane delivered in 100% oxygenSevoflurane delivered in 100% oxygenSevoflurane delivered in 100% oxygen**Locoregional anaesthesia**US Guided RBB **Ropivacaine**
**0.1 mL × Skull Length = 0.9 mL**
**Eq to 0.16 mL/kg**US Guided RBB **Ropivacaine**
**0.1 mL × Skull Length = 0.85 mL**
**Eq to 0.11 mL/kg**US Guided RBB **Ropivacaine**
**0.1 mL × Skull Length = 1.1 mL**
**Eq to 0.03 mL/kg**US Guided RBB **Ropivacaine**
**0.1 mL × Skull Length = 1 mL**
**Eq to 0.07 mL/kg**US Guided RBB **Ropivacaine**
**0.1 mL × Skull Length = 0.67 mL**
**Eq to 0.05 mL/kg**US Guided RBB **Ropivacaine**
**0.1 mL × Skull Length = 0.65 mL**
**Eq to 0.15 mL/kg**US Guided RBB **Ropivacaine**
**0.1 mL × Skull Length = 1.4 mL**
**Eq to 0.02 mL/kg**US Guided RBB **Ropivacaine****0.1 mL × Skull Length = 1 mL**
**Eq to 0.04 mL/kg**US Guided RBB **Ropivacaine**
**0.1 mL/kg = 4 mL**US Guided RBB **Ropivacaine**
**0.1 mL/kg = 2.1 mL**US Guided RBB **Ropivacaine****0.1 mL/kg = 0.73 mL**US Guided RBB **Ropivacaine**
**0.1 mL/kg = 3 mL**US Guided RBB **Ropivacaine**
**0.1 mL/kg = 0.76 mL**US Guided RBB **Ropivacaine**
**0.1 mL/kg =2.7 mL****Intra-op medication**NoneNoneNoneNoneNoneFentanyl (2 mcg/kg) IV Ketamine (0.5 mg/kg) IVNoneMedetomidine (0.002 mg/kg) IVNoneNoneKetamine (0.5 mg/kg) IVNoneNoneNone**Post-op medication**Meloxicam IV Cefalexin POMeloxicam IV Cefalexin POMeloxicam IV Cefalexin POMeloxicam IV Cefalexin POMeloxicam IV Cefalexin POMeloxicam IV Cefalexin POMeloxicam IV Cefalexin POMeloxicam IV Cefalexin POMeloxicam IV Cefalexin POMeloxicam IV Cefalexin POMaropitant IV Meloxicam IV Cefalexine POMeloxicam IV Cefalexin POMaropitant IV Meloxicam IV Cefalexine POMeloxicam PO Cefalexine PO Brinzolamide drops OS**Pain score 1 h post recovery**4/244/241/244/241/242/242/243/243/242/243/240/241/241/24**Pain score 6 h****post RBB**2/243/241/244/244/243/243/242/242/242/243/240/243/243/24**Pain score at discharge**2/243/241/24/3/24/2/242/24/1/243/240/24//

After premedication, once a good level of sedation was achieved, pre-oxygenation was provided for 5 min through a face mask at 2–4 L/min. Once general anesthesia was induced, the trachea was intubated using an appropriately sized cuffed endotracheal tube, and the cuff was inflated until no leaks were noted when an airway pressure of 20 cmH_2_O was applied by manual ventilation. The endotracheal tube was connected to the anesthetic breathing system (circle or non-rebreathing, depending on the size of the dog), and 100% oxygen was administered. Inhalant anesthesia with sevoflurane was used in 13 patients, and one case received total intravenous anesthesia (TIVA) with propofol (PropoFlo^TM^ Plus, Abbott, Berkshire, UK) for maintenance of general anesthesia.

All dogs were allowed to breathe spontaneously. Mechanical ventilation was started intraoperatively if the end-tidal carbon dioxide (PE’CO2) remained above 60 mmHg (8 kPa). Then, 20 mg/kg of cefuroxime (Zinacef, Glaxo-SmithKline Ltd., Stevenage, United Kingdom) was administered IV within 20 min prior to the surgical incision, and Hartmann’s solution (Aqupharm-11, Animal Care, York, UK) was administered at 5 mL/kg/h. Once a stable surgical plane of anesthesia was achieved, the patient was placed in lateral recumbency with the eye being enucleated uppermost.

The area around the affected eye and a small area of the lateral canthus of the eye up to the zygomatic process of the temporal bone were clipped (Figure 1a) and aseptically prepared using a 10% iodine solution within the conjunctival sac, allowing a 3-min contact time before a final flush with saline.

### 2.3. Retrobulbar Block

A supra-temporal approach [16] with US guidance was used to perform the RBB with a 12 Hz linear transducer connected to a portable ultrasound device (Philips, CX50). Sterile ocular gel was applied to the globe with the transducer positioned over the cornea and the marker directed laterally. The orientation was initially perpendicular to the transverse plane and parallel to the dorsal plane of the globe, and then it was oriented with a gentle angulation of 30–45° relative to the dorsal plane, depending on the orbital anatomy. This allowed for positioning the US marker toward the junction between the zygomatic arch and the orbital ligament. The depth and focus of the US marker were adjusted to achieve a good view of the retrobulbar space, the cone, and the optic nerve. The retrobulbar compartment was identified as a conic shape, with the extraocular muscles localized caudally to the eyeball (Figure 2a). A 22-gauge, 50 mm ultrasound needle with extension tubing (USB 50 EVOLUTION, Temena GmbH, Felsberg, Germany) was introduced using an in-plane supra-temporal approach through the temporal muscle. The site for the needle insertion was perpendicular to the skin and oriented latero-medially behind the orbital ligament, dorsal to the zygomatic process of the temporal bone, and ventral to the frontal bone (Figure 1b). When the tip and part of the needle were identified ultrasonographically, the needle was advanced latero-medially by crossing the extraocular muscles to reach the intraconal space (Figure 2b). Before injection of the local anaesthetic (LA), color flow Doppler was used to localize the ophthalmic artery, and suction was performed to ensure negative aspiration of blood. During injection, it was also confirmed that no resistance was noted upon injection to avoid injecting iside the optic nerve sheath. In 6 out of 14 dogs, the injection was performed with 0.1 mL/kg of ropivacaine (Naropin, 0.75%, Aspen Pharma, Maidenhead, United Kingdom), as reported by Chiavaccini et al. 2017 [16]. For the remaining eight dogs, the technique was the same, but the dosage of ropivacaine was calculated in relation to the cranial length (Lcr) as described by Klaumann et al. 2018 [17] with the formula of 0.1 mL/cm Lcr. During the injection, the distribution of LA was observed with the US marker as an anechoic shadow inside the retrobulbar cone with an enlargement of it (Figure 2c).

### 2.4. Surgery

After the execution of the block, the eye was cleaned of any remaining gel and received final preparation with a diluted iodine solution. The dogs were then transferred to the operating room. The start of surgery was at least 15 min after the block. A transconjunctival technique for enucleation was performed in 13 dogs, as well as a transpalpebral approach in one patient (case 5).

In addition, possible intraoperative complications were managed and treated as required.

In cases of intraoperative uncontrolled hemorrhaging, 10 mg/kg of intravenous tranexamic acid (Tranexamic acid, Sandoz, Camberley, UK) was administered, and at the end of the surgery, a gauze was temporarily sutured tightly on top of the surgical wound.

All the surgical procedures were performed by either a board-certified ophthalmologist or a resident of the European College of Veterinary Ophthalmologists under the supervision of a board-certified ophthalmologist.

### 2.5. Monitoring

The physiological variables during anesthesia were recorded every 5 min by either an anesthesia intern or nurse and included the heart rate (HR), respiratory rate (fR), noninvasive oscillometric systolic, diastolic, and mean arterial blood pressures (SAP, DAP, and MAP, respectively), arterial hemoglobin oxygen saturation with pulse oximetry (SpO2), end-tidal carbon dioxide partial pressure (EtCO2), end-tidal sevoflurane concentration (EtSevo), electrocardiography (ECG), and esophageal temperature using a multiparameter monitor (Datex-Ohmeda, S/5, GE Healthcare, Cheshire, UK).

The HR, *f*R, and MAP were recorded 5 min before the surgical incision were considered as baseline values for the purpose of evaluating the nociceptive responses. If at the time of surgical incision or at any point during surgery one or more of these parameters exceeded the baseline values by 20%, rescue of analgesia was administered immediately in the form of an IV bolus of 0.5 mg/kg ketamine (Anesketin, Dechra, Shrewsbury, UK) or 0.002 mg/kg fentanyl (Fentadon, Dechra, Shrewsbury, UK).

### 2.6. Recovery and Postoperative Plan

Before recovery from general anesthesia, the bladder was expressed to ensure no urinary discomfort during the recovery period. Eleven patients received meloxicam (Metacam, Boehringer Ingelheim Vetmedica GmbH, Rohrdorf, Germany) IV at an initial dose of 0.2 mg/kg once recovered from general anesthesia. The remaining three dogs had previously received meloxicam the day before surgery, and the treatment was continued PO at the maintenance dose of 0.1 mg/kg.

The Short Form of the Glasgow Composite Pain Scale (CMPS-SF) was used to evaluate the postoperative pain in all dogs [18]. The pain was scored at 1 h following recovery from general anesthesia, 6 h after the RBB, and before the discharge of the patient. Additional pain scoring was performed if the patient was considered unsettled or uncomfortable in the kennel up until the time of discharge. The postoperative rescue analgesia consisted of 0.2 mg/kg methadone if the pain score was ≥6 out of 24 and no further treatment if it was ≤5 out of 24.

Two hours after the end of surgery, all patients were checked to rule out possible active bleeding. Water and a small amount of food were offered as soon as the patients were fully recovered from anesthesia. Three out of 13 dogs refused the food, while the remainder ate 100 % of the food offered.

The dogs were discharged from the hospital approximately 6 h after full recovery from general anesthesia with the following medication to continue at home: meloxicam (0.1 mg/kg PO once daily for 5 days) and 15 mg/kg of cefalexin PO (Cephacare, Animal Care, York, UK) twice a day for 7 days. The owners were instructed to recognize any signs of discomfort or pain and to contact the clinic if there were complications or behavioral changes (e.g., apathy or anorexia) at home.

## 3. Results

The most common reasons for enucleation in this case series were intractable glaucoma and end-stage ocular disease. Case 12 required additional preoperative sedation with 0.005 mg/kg of dexmedetomidine IM (Sedadex, Dechra, Shrewsbury, UK) due to insufficient effectiveness of the initial dose. In case 8, an additional intravenous bolus of 0.002 mg/kg of medetomidine (Sedator, Dechra, Shrewsbury, UK) was administered 10 min after induction of general anesthesia due to an insufficient depth of anesthesia before moving to the theater. Case 11 required an additional bolus of 0.5 mg/kg of alfaxalone (Alfaxan, Juorx, Crawley, UK) IV 5 min after induction to deepen the plane of anesthesia. Only 2 out of 14 patients required intraoperative rescue analgesia. Case 6 received 0.5 mg/kg of ketamine IV (Anesketin, Dechra, Shrewsbury, UK) at the time of optic nerve transection due to an increase in *f*R greater than 20% compared with the baseline value. Moreover, a second rescue analgesic dose of 0.002 mg/kg of fentanyl (Fentadon, Dechra, Shrewsbury, UK) was also required due to the persistence of a high *f*R level. In case 11, a single dose of 0.5 mg/kg of intravenous ketamine (Anesketin, Dechra, Shrewsbury, UK) was administered at the end of surgery after skin closure due to a greater than 20% increase in blood pressure. Case 7 had an increase of 15% in both HR and *f*R during manipulation of the eyelids, but no rescue analgesia was necessary. No episodes of vomiting or regurgitation, excessive bleeding, hypotension, or oculo-cardiac reflex were recorded in any patient during general anesthesia. Recovery from general anesthesia was uneventful in all patients. The postoperative pain scores using the CMPS-SF are reported in detail in Table 1. No rescue analgesia was required in any dogs during the whole postoperative period. All dogs were discharged from the hospital without any concern, and none of the owners contacted the clinic for rescue analgesia or complications.

## 4. Discussion

This retrospective case series shows the feasibility of OFA in 13 out of 14 patients undergoing unilateral enucleation using a multimodal analgesic regime, highlighting the importance and reliability of loco-regional anesthesia. The combination of a retrobulbar block with the alpha 2 adrenergic agonist, a non-competitive NMDA antagonist, and NSAIDs was effective at preventing intraoperative nociception in 12 out of 14 cases and provided good postoperative pain relief in all 14 cases postoperatively. This represents a typical drug combination used during OFA in both human and veterinary medicine [19].

In the veterinary literature, OFA regimes have already been described in case reports of soft tissue [10,20,21,22,23,24], orthopedics [25] and neurosurgeries [26,27], highlighting the importance of loco-regional anesthesia in the context of multimodal analgesia.

Despite systemic opioids providing excellent perioperative analgesia and that opioid-induced side effects at licensed doses do not have a negative clinical impact [5], other alternatives could be investigated in the management of perioperative analgesia of patients undergoing enucleation.

Although the evidence regarding side effects in dogs is not as clear and exhaustive as in human medicine, OFA can eliminate the presence of opioid-induced nausea and vomiting [28,29,30], panting [31], and vocalization [32,33] and reduce the risk of opioid-related intraoperative gastro-esophageal reflux (GOR) [34], postoperative ileus [35,36], urinary retention [37,38], and the impairment of tear production [39].

The latter is of particular importance for the safety of the remaining eye after unilateral enucleation.

The goal of an OFA protocol is to avoid unnecessary perioperative opioid usage and limit their use only when needed as intraoperative or postoperative rescue analgesia, with the aim to improve the quality of recovery from general anesthesia [40].

The use of US guidance for the supratemporal RBB has been recently described by Briley et al. (2023) [41], and as for other peripheral nerve blocks, it can significantly improve the success rate and reduce complications [42]. This is especially relevant for retrobulbar blocks, where is not possible to benefit from nerve stimulation.

Various other approaches for the RBB have been described to provide analgesia during enucleation surgery [43]. Among these techniques, the most commonly used one is the one described by Accola [44], using a bent 22-gauge spinal needle inserted blindly through the inferior eyelid. Other approaches such as the peribulbar block (PBB) have been described because of the advantage of keeping the needle away from the globe. However, higher volumes of LA are required. Recently, a US-guided peribulbar block with the temporal approach was described in dog cadavers by Foster [45], using a volume of 0.2 mL/cm Lcr. A remarkable and potentially dangerous result of this study was the intracranial spread of contrast noted following 5 out of 10 (50%) injections.

The US-guided RBB with the supra-temporal approach is routinely used at our practice because it prevents complications such as corneal injury and reduces the risk of globe penetration (adequate distance from the needle to the globe) and retrobulbar hemorrhage by identifying the ophthalmic artery using color flow Doppler. Moreover, US guidance allows one to follow the needle during the procedure, reducing the risk of accidental injection within the optic nerve sheath and subsequent brainstem anesthesia. An additional advantage of the RBB is that dogs undergoing enucleation surgery have significantly lower odds of developing an oculo-cardiac reflex [46].

Aside from that, the RBB performed for enucleation surgery in dogs seems to not increase surgical or anesthetic complications and may improve the quality of recovery from general anesthesia [47].

Concerning the distribution of LA, both of the adopted volumes have been reported to be enough to fill the intraconal space [16,17]. However, according to Chiavaccini et al. [16], a volume of 0.1 mL/kg was reported to also distribute to the extraconal space in 50% of the cadavers through the possible lack of an intermuscular septum between the extraocular muscles [16]. This further distribution allows a blockade of the nerves that provide sensory innervation to the eyelids, but this may not happen in 100% of the animals.

According to this, in large dog breeds, when the calculated volume is based on the Lcr instead of the body weight, the difference could become relevant for LA distribution. This could have been the situation in case 7, a 53 kg Great Dane who received a volume of ropivacaine of 1.4 mL, considering the length of the skull. However, the calculation based on the dog’s body weight would have been 5.3 mL. It is likely that the small volume used in this case was not enough to allow the spread into the peribulbar space, and this could explain the increase in HR and *f*R observed during eyelid manipulation. Nonetheless, this cannot explain the situation in case 6, a 4.2 kg Chihuahua that received a total volume of ropivacaine of 0.65 mL based on the length of skull. However, the volume based on the body weight would have been only 0.23 mL. This case was not entirely opioid-free, as it received a bolus of fentanyl intraoperatively after receiving a previous ketamine bolus. It is likely that in this case, the distribution of the LA was not adequate to completely desensitize the intra- and extraconal structures, but the reason for this failure remains unknown. Such case shows that although an OFA was planned, it was necessary to administer an opioid for the management of acute nociception activation intra-operatively. This emphasizes the rational use of opioids, administered to effect only when needed.

In case 11, a 7.3 kg Maltese crossed with a Poodle, no reaction was detected during the surgical procedure. However, at the end of surgery, when no surgical manipulation was occurring, the *f*R increased, and a rescue ketamine bolus was administered by the nurse in charge of the anesthesia. After reviewing the anesthetic record, the authors consider this reaction to be related to a possible lightening of the anesthetic depth.

The Short Form of the Glasgow Composite Measure Pain Scale (CMPS-SF) was selected for pain evaluation in the postoperative period [18]. Pain was evaluated 1 h after recovery from general anesthesia and thereafter at 6 h following the RBB according to Wagatsuma [48], who described that the duration of the sensory blockade after PBB with a ropivacaine injection was approximately 6 h. The last pain evaluation was performed again prior to discharge to ensure that the patients did not need any further analgesia before going home.

Certainly, enucleation itself represents a relief in patients suffering from severe glaucoma or perforated corneal lesions, and therefore, removing the eyeball may improve their comfort level.

Despite the excellent analgesic effect on the postoperative period observed in this case series, randomized blind prospective clinical trials are needed to confirm and strengthen the clinical validity of the reported retrobulbar anesthetic technique. Particularly interesting would be the comparison between the loco-regional anesthesia technique adopted in this case series and traditional systemic opioid administration, as well as with other periocular anesthesia techniques in dogs undergoing enucleation.

Recently, in a study on the comparison of a blind RBB and US-guided supratemporal RBB in dogs undergoing enucleation, the US-guided supratemporal RBB was more effective at decreasing intraoperative opioid analgesia than the blind technique [41].

Several limitations are present in this case series, such as the differences in the anesthetic protocols with dogs receiving the alpha 2 agonist in premedication and only some receiving ketamine for co-induction. Both of these drugs having some analgesic action.

Furthermore, the use of a single reading of measurements for the HR, RR, and BP as an assessment of intraoperative nociception [49] could have been influenced by the presence of the alpha 2 agonist, where it would be expected for the HR to gradually increase with time.

Another limitation consists of the different methods for calculating the volume of the LA. This is due to the fact that during the clinical activity in that period, new approaches were proposed over time according to the scientific literature of the moment.

Concerning the pain score, the modified Glasgow Composite Pain Scale is not a specifically validated pain scale for ophthalmic surgery. However, according to the authors’ knowledge, there are currently no validated ophthalmic pain scales in dogs.

Nevertheless, all evaluators had extensive experience assessing ophthalmic pain, and in our opinion, preoperative pain due to intraocular diseases needs to be differentiated from pain following enucleation surgery. In fact, the latter relates more to the extraocular surrounding structures rather than the eye itself, which has already been removed.

Some patients did not have a pain score at the time of discharge, and the dogs were discharged at different times from recovery of general anesthesia. This is due to the retrospective nature of this case series, where a well-designed 24-h pain score was not present.

Certainly, the feasibility of OFA protocols depends on several factors, such as the personnel training, availability of equipment required to perform the anesthetic blocks, and capability to perform postoperative periodic pain assessments by qualified people. Moreover, OFA may not always be feasible due to possible comorbidities that require the avoidance of analgesic drugs, such as alpha 2 agonists or ketamine (i.e., certain cardiac pathologies).

To the authors’ knowledge, there are no previous case reports on OFA for enucleation surgery in veterinary medicine. The authors consider the reporting of this case series important for discussing alternative analgesic approaches, with the goal of promoting an individualized pain management focus on the rule of loco-regional anesthesia in the context of multimodal analgesia.

## 5. Conclusions

In this retrospective case series, the ultrasound-guided retrobulbar block in an OFA regime was effective at managing the anesthesia and perioperative analgesia of 13 out of 14 patients undergoing enucleation. However, randomized blinded prospective clinical trials are needed to confirm the clinical efficacy and potential advantages of this loco-regional anesthesia technique in comparison with classical systemic analgesia and other periocular anesthesia techniques.

## Figures and Tables

**Figure 1 animals-13-02059-f001:**
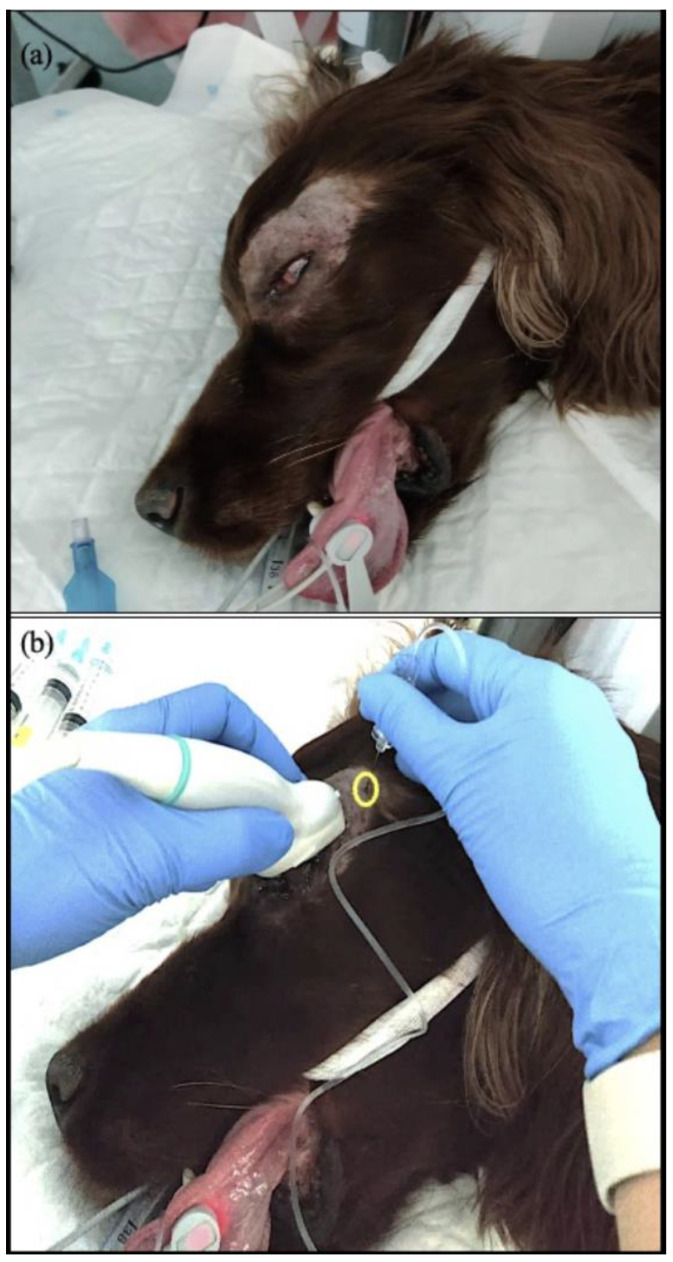
Preparation and execution of the US-guided retrobulbar block. (**a**) Area shows the area around the affected eye and a small area of the lateral canthus of the eye up to the zygomatic process of the temporal bone clipped and prepared aseptically. (**b**) Transducer position on the cornea and upper eyelid parallel to the eye equator with the marker oriented laterally. The site of the needle insertion (yellow ring) caudal to the supraorbital ligament, dorsal to the zygomatic process of the temporal bone, and ventral to the frontal bone.

**Figure 2 animals-13-02059-f002:**
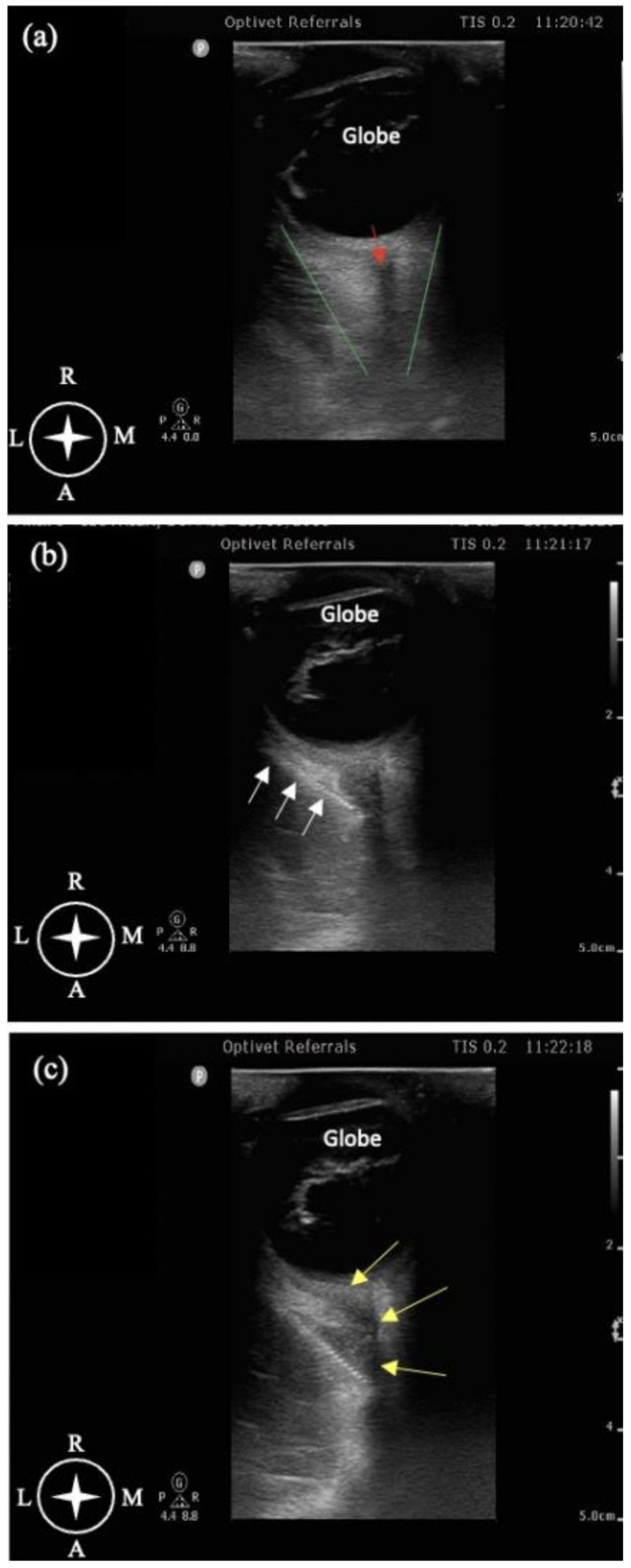
Ultrasound view of retrobulbar block. (**a**) Ultrasonographic anatomy of the eyeball. The red arrow indicates the optic nerve running inside the retrobulbar cone, identified as the area within the green lines. (**b**) Needle (white arrows) directed inside the retrobulbar cone. (**c**) LA spread inside the retrobulbar cone. LA appears as an anechogenic shadow inside the retrobulbar cone, indicated by the yellow arrows. R = rostral; L = lateral; M = medial; A = aboral.

## Data Availability

The presented case series is not a study, and no data were collected.

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
