# Peer review of "The Outcomes of an Opioid-Free Anaesthetic Plan in Fourteen Dogs Undergoing Enucleation Using an Ultrasound-Guided Supra-Temporal Retrobulbar Block: A Retrospective Case Series"

_animals, 2023, doi:10.3390/ani13132059_

Round 1
Reviewer 1 Report
Thank you for this interesting paper, I really like the issue and I agree with authors that an excellent analgesia can be achieved without the need of using opioids. I also think they are not so awful, but a rational use is indispensable in our field. The explain why they decided to write this paper and the aim of it. There is no published serial cases as this one and it is suitable for the journal and the current edition is it aimed to be published in. I think it needs some improvement before acceptance and I recommend a good revision before the acceptance of this promising paper. Thank you for this paper and please, feel free to answer my questions/recommendations as you estimate opportune.
Page 1, line 18: “tenporal”? Please correct for “temporal”
Page 1, abstract. You speak about locoregional anaesthesia, but you don’t state which block you use. Abstract is really general, even more than simple summary.
Page 1, line 42: opioids are reversal or antagonized? I think it is not the same.
Page 3, table 1: it is not correctly in line the arrows. Maybe it was a problem from converting it to pdf.
Page 4, line 126: how long before surgery was omeprazole given?
Page 4, table 2: please, check if you need to use ® at naming the branches drugs. Why did you mix branches with active drugs in your list? I find it confusing, but maybe you had a good reason for it. I would try to stick to one option
Page 7,lines 135-137: you wrote that after IM premedication, catheter was placed. But some animals received IV premedication. Please clarify what you did with these animals. Because it looks all animals were premedicated IM.
Page 7, line 150: I agree with you that we use jaw tone to check depth of anaesthesia. But some people consider that it is a really subjective indicator and it is difficult to monitor it. I find strange that you changed the position of the animal when the animal lost palpebral reflexes and jaw tone, is not that the moment of the anaesthesia induction and tracheal intubation? Did the have palpebral reflex and jaw tone after intubation? I would consider to delate that sentence and just say that the animal was placed on lateral recumbency.
Page 11, line 206: How long did it take from the premedication to start the surgery?
Page 12, line 291: Please, check decimals. In English, “0.002” instead of “0,002”
Page 12, line 298: if you used fentanyl you can’t state you did OFA in all these animals. Therefore, your title is incorrect, I would change it to stick to your reality.
Page 12, line 300: I think you forgot to erase all the revision that you did on word: “0.5 mg/kg IV ketamine (Anesketin, Dechra) 0.5 mg/kg IV”
Page 13, lines 310-316: I am sorry to say this, but you can’t state all these cases were OFA if you used intraoperative fentanyl bolus in one. You need to re-write this.
Page 13, line 316: “rapresents” means “represents”?
Page 13, line 345: there is a lack of space: “previously[17,18]”
Page 14, lines 376-378: “to avoid injection within the optic nerve sheath with subsequent brainstem anaesthesia.” I think you should be less ‘absolute’ with this sentence. It is true, it helps to avoid these, but it doesn’t mean it can’t happen at all. If you have a citation that probe this, please cite it.
Page 15, lines 450-452: “This is due to the fact that during the clinical activity in that period new approaches were proposed over time according to the scientific literature of the moment. One of them is the calculation of the volume of local anesthetic according to the length of the skull.” The second sentence it’s a repetition of what you have already said on your paper. I think you should focus in discussing advantages/disadvantages of both approach and the literature that support them. You already said that length of skull might underestimate volume for large dogs and overestimate volume for tiny dogs.
Page 15, lines 461-462: Did you say the time between the block and the discharge? A block that last at the best 6h and not giving any extra analgesia for home, looks estrange. Do you think a dose of NSAIDS is enough? Did the animal keep this therapy? Were owners instructed to recognize pain in their dogs?
Page 15, line 465: Is not “canonical” a little bit subjective word? I don’t think it is appropriated for a scientific paper.
Page 15, lines 461-465: Do not belong together these two sentences?
Page 16, line 504: I find funny that you hided the name of the hospital at materials and methods, but you stated here and on the US images. Did the journal asked you to hide it at materials and methods? Because it is clear where the study was performed.
General comments that need to be check:
- The title is not true if one dog received fentanyl, this is not OFA.
- The abstract did not say the only block here discussed.
- The introduction is long and part of the information could be move to the discussion, where it could improve the discussion part without repeating things.
- I am sorry, but the conclusion, title, introduction and promoting this paper as a description of OFA is not really true if we consider that you gave fentanyl to one dog. I think you should omit this case or you need to remove OFA from title, abstracts and conclusions. You need to say that in one case you used fentanyl and please, explain and discuss why this was done during the discussion. I understand a nurse or an intern did it, because of indicators of nociception, but it really destroys the OFA defense and you barely comment about it. You focused more on the ketamine bolus given because of light anaesthesia.
- I find the discussion a little bit disjointed and it need to improve it, because I think you really can do it.
- Please, check how you wrote your references. It gives an impression of careless. Did anyone check they were correctly written? Some of them lack information, spaces between numbers and the name of the journal, authors, one is written the journal as (internet), a couple are written in capital letters… I know endnote or other programs do this, but you need to be really careful about checking them. You wrote an interesting case report and after reading the references it looks like you did it in hurry up because you wanted to submit something.
Thank you again for your papers. I found it really interesting and I am sure it would be great after some revisions.
Reviewer 2 Report
I have some questions:
- Do you usually administer omeprazole at 1 mg/kg in all animals go to a surgery?
- You mention that pain evaluation was 1 hr post following recovery from general anaesthesia. In my opinion, at this time, it is possible that the animal show sedation signs because of the proximity of general anaesthesia, masking the pain. It could be a limitation.
- You mention that the dogs were discharged from the hospital approximately 6 hrs after full recovery from general anaesthesia (line 284). In table 2, in 9 dogs, there is a pain evaluation at 6 hrs post block and prior to discharge. So, do both measurements agree? I think you would have to mention about what time they are discharged from the end of general anaesthesia.
I suggest some minor changes:
- line 76: ..."often resulting in blind and painful eyes regractory to..."
- line 316: "This represents..."
- line 329: In my opinion, this paragraph should join to previous one.
- line 423: In my opinion, this paragraph should join to previous one.
- line 429: you should consistent with this word: loco-regional or locoregional?
- line 443: In my opinion, this paragraph should join to previous one.
- line 450: In my opinion, this paragraph should join to previous one.
- line 464: In my opinion, this paragraph should join to previous one.
- Line 481: "Learning an OFA regime..."
- line 492: "... in OFA regime..."
- Figure 2: only in this figure, you name local anaesthetic as LA. You don´t make that mention throughout the manuscript.
Reviewer 3 Report
Dear colleagues (authors and editors),
It is my pleasure to review the paper entitled: “Opioid free anaesthesia in fourteen dogs undergoing enucleation using an ultrasound-guided supra-temporal retrobulbar block” with the manuscript ID: animals-2413599. This work addresses an interesting topic of timely relevance to the field of veterinary anaesthesia and analgesia. Although opioids are widely used, they are not without complications. Moreover, opioids are subject to numerous restrictions in many countries. For these reasons, opioid-free anaesthesia (OFA) represents an emergent strategy with a huge interest because of the possibility to reduce the use of these drugs. The objective of the present study was to evaluate the clinical feasibility of OFA in dogs undergoing unilateral enucleation surgery. Authors provided a multimodal perioperative analgesia with the focus on locoregional anaesthesia using an ultrasound-guided retrobulbar block (RBB). Locoregional anaesthesia in veterinary medicine is another hot topic. New approaches are published every day to block very precisely certain nerve structures relevant in the clinical scenario. RBB have been described by other authors previously, both blindly and ultrasound-guided, but in this paper this locoregional block was used in animals that did not receive opioids perioperatively, which is a novelty to my best knowledge.
Authors investigate retrospectively a total of 14 clinical cases of dogs undergoing unilateral enucleation in which an OFA protocol was used, including as analgesic drugs dexmedetomidine and/or ketamine, along with an ultrasound-guided RBB. According to authors, 2 out of 14 dogs did require rescue analgesia intraoperatively, and none needed rescue analgesia during de postoperative period. Authors conclude that ultrasound-guided RBB in an OFA regime was effective in managing the anaesthesia and perioperative analgesia in dogs undergoing enucleation.
The present work is a clinical case series, with a relatively small number of animals, where a group of dogs are exposed to the intervention (OFA protocol with ultrasound-guided RBB), without a control group. It produces a low level of evidence according to the evidence-based medicine guidelines. Consequently, any conclusions drawn from this work should be treated with caution, and the reader should keep in mind the nature of this research. Along discussion section, authors identify and justify several of the limitations of this study.
After a thorough reading of this manuscript there are several suggestions that I would like to share with authors before publication:
1. Regarding the structure of the manuscript, authors divided it in: 1. Introduction; 2. Cases Presentation; 3. Treatment; 4. Discussion; and 5. Conclusion. I would propose a more classical structure, with 1. Introduction; 2. Methodology (or Materials and Methods), where authors could include different sub-sections like data retrieving (including inclusion / exclusion criteria), pre-anaesthetic procedure, retrobulbar block (figures), intra-operative monitoring, surgery, pain score evaluation, and recovery and post-operative plan; 3. Results, where author describe any observation related with this study like signalment and medical conditions, individual case description (table), analgesic rescue requirements, and complications (if any) [basically it is the former 3.6. Outcome section in manuscript]; 4. Discussion; and 5. Conclusions.
2. I would recommend authors to include some reference/s to hold the paragraph between lines 75 and 81.
3. In my opinion, paragraph between lines 85 and 99 in which authors explain thoroughly innervation of retrobulbar area is too deep for this paper, and I would simplify considerably or remove it. If so, reference related with this content would be removed (16).
4. In the last paragraph of the introduction, the sentence “The most common reasons for enucleation were intractable glaucoma and end-stage ocular disease” is an observation derived from the study, and I would remove it from the introduction and paste in the Results section.
5. The last paragraph in the introduction should clearly state the hypothesis and objective/s of this study.
6. Table 1 results a bit redundant for me. The first column of table 2 already basically summarises the data from table 1. I strongly recommend reediting Table 2 to make if more friendly. Maybe some column could be combined or summarised. Pain score column is crucial, and I consider it would be subdivided in three columns: 1h post-recovery, 6h post-RBB, and prior to discharge. In my opinion, table would be clearer to reader horizontally. I would change any commercial name in the table to the name of the drug, i.e., change “Cephacare” for “cefalexin”.
7. I would consider adding a short video about the ultrasound guided RBB as a supplementary material. It would be of great value to help reader to understand this recently described clinical procedure.
8. I consider that surgery paragraph (from line 206 to 234) is too extensive for the scope of this paper. I strongly recommend summarizing and hold it with relevant reference/s. I would clarify that “all surgical procedures were performed by a board-certified ophthalmologist or under the supervision of a board-certified ophthalmologist”.
9. I would consider putting the explanation about intraoperative haemostasis (lines 251 to 254) in “surgery” section, instead of in “monitoring” section. I would also propose to remove the paragraph between lines 255 to 265, mainly because no complication related to severe haemorrhage or oculo-cardiac reflex were described in this case series. If so, references related with this content would be removed (19, 20, 21).
10. The authors have done a thorough job in the discussion, but I consider it too long. It is more than three pages, and I would consider reducing at least one page. I would propose to try to reduce and summarise the content of this part of the manuscript and adapt it to the nature of the paper (case series).
11. I would consider joint in a sole paragraph from line 322 to 333.
12. Authors state in lines 337 and 338 that “This US-guided supratemporal RBB is an emergent technique as it has not been described previously”. In a recent accepted paper, Briley et al described a similar approach (Briley JD, Washington D, Westermeyer HD, Posner LP, Chiavaccini L. Comparison of a blind and an ultrasound-guided technique for retrobulbar anesthesia in dogs undergoing unilateral subconjunctival enucleation. Vet Ophthalmol. 2023; 00:1-7. doi:10.1111/vop.13092). Please, consider this publication and reedit the text in the manuscript where appropriate.
13. I would consider joint lines 380 to 382 to the previous paragraph.
14. There is a recent work by Bartholomew et al., where authors investigate complications related to retrobulbar block in dogs undergoing unilateral enucleation surgery. I should consider including this work along discussion where appropriate (Bartholomew KJ, Smith LJ, Bentley, Lasarev MR. Retrospective analysis of complications associated with retrobulbar bupivacaine in dogs undergoing enucleation surgery. Vet Anaesth Analg 2020,47,588-594. doi.org/10.1016/j.vaa.2020.04.007).
15. I would consider summarize paragraph between lines 383 to 406.
16. I would consider joint lines 413 and 414 to the next paragraph.
17. I suggest including reference number 23 in line 414.
18. In line 415, please define “GA” (if it is “general anaesthesia”, it should be properly defined before).
19. I would consider joint lines 443 to 446 to the previous paragraph.
20. I would consider joint in a sole paragraph from line 447 to 452.
21. I would consider joint in a sole paragraph from line 461 to 465.
22. In general, authors should adhere strictly to the “Instruction for Authors” and prepare reference list as recommended by the ACS style guide. I would encourage authors to reduce the number of references.
- Reference 6 (doi: 10.1113/jphysiol.1987.sp016731): I would consider removing it. This is a very old reference, and I believe it could be unnecessary.
- Reference 12: Please, consider that this reference has been published in Pain Physician 2008; 11:S105-S120.
- Reference 26 (doi: 10.1002/j.1552-4604.1993.tb05618.x): I would consider to removing it. This is a very old reference, and I believe it could be unnecessary.
- Reference 35 (doi: 10.1016/0014-2999(81)90408-8): I would consider removing it. This is a very old reference, and I believe it could be unnecessary.
- Reference 36: Please, revise the reference (maybe it is doi: 10.1023/a:1015081822336).
- Reference 41: Please, consider that this reference has been published in Current Opinion in Anaesthesiology 2006, 19:630–639.
- Reference 47 is an abstract presented at the American College of Veterinary Anesthesia and Analgesia Annual Meeting on September 12, 2021, Nashville, TN, (USA). I would suggest replacing it by the recently published paper: Briley JD, Washington D, Westermeyer HD, Posner LP, Chiavaccini L. Comparison of a blind and an ultrasound-guided technique for retrobulbar anesthesia in dogs undergoing unilateral subconjunctival enucleation. Vet Ophthalmol. 2023; 00:1-7. doi:10.1111/vop.13092.
- Reference 51: Please, consider that this reference has been published in Canadian Veterinary Journal 2020 Jun;61(6):621-628.
23. I finally suggest some minor details like:
- In line 18, please, change “tenporal” by “temporal”.
- In line 19, I would change the word “monolateral” by “unilateral”.
- In line 112, author say “…three-month period from March to December 2020”, but from March to December there are 10 months. Could you clarify that, please?
- In the footnote of Figure 2 (line 201), I would suggest changing the term “aboral” for “caudal”. Aboral is a term mainly used for gastrointestinal tract.
- In line 316, please, change “rapresents” by “represents”.
